# Coronary calcium scoring assessed on native screening chest CT imaging as predictor for outcome in COVID-19: An analysis of a hospitalized German cohort

Gregor S. Zimmermann[1]☯*, Alexander A. Fingerle[2]☯, Christina Müller-Leisse[2], Felix Gassert[2], Claudio E. von Schacky[2], Tareq Ibrahim[1], Karl-Ludwig Laugwitz[1], Fabian Geisler[3], Christoph Spinner[3], Bernhard Haller[4], Markus R. Makowski[2], Jonathan Nadjiri[2]

**1** Department of Internal Medicine I, School of Medicine & Klinikum rechts der Isar, Technical University of Munich, Munich, Germany, **2** Department of Diagnostic and Interventional Radiology, School of Medicine & Klinikum rechts der Isar, Technical University of Munich, Munich, Germany, **3** Department of Internal Medicine II, School of Medicine & Klinikum rechts der Isar, Technical University of Munich, Munich, Germany, **4** Institute of Medical Informatics, Statistics and Epidemiology, Technical University of Munich, Munich, Germany

☯ These authors contributed equally to this work.
* gregor.zimmermann@tum.de

**Data Availability Statement:** We uploaded a minimal anonymized data set as Supporting information file. Following local laws and Clinical

## Abstract

### Background

Since the outbreak of the COVID-19 pandemic, a number of risk factors for a poor outcome have been identified. Thereby, cardiovascular comorbidity has a major impact on mortality. We investigated whether coronary calcification as a marker for coronary artery disease (CAD) is appropriate for risk prediction in COVID-19.

### Methods

Hospitalized patients with COVID-19 (n = 109) were analyzed regarding clinical outcome after native computed tomography (CT) imaging for COVID-19 screening. CAC (coronary calcium score) and clinical outcome (need for intensive care treatment or death) data were calculated following a standardized protocol. We defined three endpoints: critical COVID-19 and transfer to ICU, fatal COVID-19 and death, composite endpoint critical and fatal COVID-19, a composite of ICU treatment and death. We evaluated the association of clinical outcome with the CAC. Patients were dichotomized by the median of CAC. Hazard ratios and odds ratios were calculated for the events death or ICU or a composite of death and ICU.

### Results

We observed significantly more events for patients with CAC above the group's median of 31 for critical outcome (HR: 1.97[1.09,3.57], p = 0.026), for fatal outcome (HR: 4.95 [1.07,22.9], p = 0.041) and the composite endpoint (HR: 2.31[1.28,4.17], p = 0.0056. Also,

Institutional Review Board (Ethic Committee of the Technical University Munich, Director: Prof. Dr. G. Schmidt) the data sharing is restricted due to potentially identifying patient information in this study. We added a excel sheet of the CAC and the endpoints of our study. An identification of an individual patient is not possible in this excel sheet. A reasonable request for a data access could be sent by researchers who meet the criteria for access to confidential data to the corresponding author only after permission from the Clinical Institutional Review Board (Director: Professor Dr. G. Schmidt, contact: ethikkommission@mri.tum.de).

**Funding:** The authors received no specific funding for this work.

**Competing interests:** The authors have declared that no competing interests exists.

**Abbreviations:** CAC, coronary calcium score; CAD, coronary artery disease; COVID-19, coronavirus disease 2019; CT, computed tomography; ICU, intensive care unit; PCR, polymerase chain reaction; SARS-CoV-2, severe acute respiratory syndrome coronavirus 2.

odds ratio was significantly increased for critical outcome (OR: 3.01 [1.37, 6.61], p = 0.01) and for fatal outcome (OR: 5.3 [1.09, 25.8], p = 0.02).

## Conclusion

The results indicate a significant association between CAC and clinical outcome in COVID-19. Our data therefore suggest that CAC might be useful in risk prediction in patients with COVID-19.

## Introduction

Severe acute respiratory syndrome coronavirus-2 (SARS-CoV-2) infection was first detected in Wuhan, China and is responsible for the widespread coronavirus disease 2019 (COVID-19) [1–3]. Since the initial detection, the SARS-CoV-2 pandemic claimed many lives due to severe respiratory and multi-organ failure. The number of cases continues to rise; yet there have been more than 50 million confirmed COVID-19 cases and more than 1.3 million deaths worldwide [4]. Initial reports from all over the world indicate a high mortality and stressed intensive care unit (ICU) capacity [5, 6].

Native chest CT showed a high sensitivity in the initial diagnosis of COVID-19 and leads further diagnostic steps [7]. Therefore, chest CT is considered as a screening tool for COVID-19 in patients with severe respiratory insufficiency. Due to rapid results in comparison to RT-PCR for SARS-CoV-2, chest CT is an integral part of the diagnostic process, especially in hospitalized patients with respiratory insufficiency. Typical CT imaging features of COVID-19 direct the suspicion without delay at an early stage after hospital admission and can enable further therapeutic regimes and preventive quarantine to be initiated promptly [8]. In addition, there are ongoing studies whether chest CT allows further conclusions regarding the degree of severity [9, 10].

Patients with cardiovascular risk factors or documented coronary artery disease could be identified as high risk subpopulations for fatal outcomes in earlier studies [10–12]. In addition, the systemic inflammatory response to COVID-19 leads to an increased incidence of cardiac events such as atrial fibrillation, congestive heart failure and coronary ischemic events [13, 14]. Elevated troponin as a laboratory finding for myocardial damage could be identified as a significant inverse prognostic marker for survival [10, 15].

Coronary artery calcification (CAC) is an established marker of coronary atherosclerosis and a simple and reliable quantification of CAC is achieved noninvasively by computed tomography. CAC is an established risk predictor for cardiovascular events since many years and provides a more accurate risk prediction for the individual patient than conventional risk factor scores. Several studies showed a significant association between future cardiovascular events and the amount of CAC [16, 17]. Moreover, the absence of CAC is associated with an excellent cardiovascular prognosis [18].

Due to limited capacities of intensive care facilities, the rapid identification of patients with an increased risk for severe outcome or need for intensive care is of high importance. This retrospective study seeks to evaluate the short-term prognostic value of CAC in patients with COVID-19 calculated from standard chest CT.

The aim of this study was to assess the prediction of clinical outcome in patients suffering from COVID-19 based on the analysis of the coronary calcium-scoring obtained from native chest CT scans during initial screening procedures in a high volume university center for the treatment of COVID-19.

## Methods

### Study population and endpoints

This single-center, retrospective, observational study was conducted at the university hospital Klinikum rechts der Isar of the Technical University Munich. All patients with laboratory confirmation of COVID-19 infection, according to the interim guidance of the World Health Organization from March 20th 2020, and native chest CT were included. Clinical information of patients diagnosed and treated, during hospitalization from March 05th to April 15th, 2020 was collected by the study team. The research protocol was approved by the local Clinical Institutional Review Board (Ethic Committee of the Technical University Munich) and complies with the 1964 declaration of Helsinki and its later amendment. Data sets were anonymized after acquisition and before processing. The local ethics committee approved acquisition, processing and publication of results from an anonymized data set. Local ethics committee waived informed consent for this particular study.

We included only hospitalized patients with severe manifestation and need for hospitalization. No outpatients with mild manifestation of COVID-19 were included. Patient records were reviewed retrospectively after anonymization and the duration of inpatient treatment, transfer to intensive care and outcome were recorded.

We defined three endpoints: critical COVID-19 and transfer to ICU, fatal COVID-19 and death, as well as a composite endpoint (critical and fatal COVID-19) comprising death and ICU treatment. Patients with hospitalization but without fatal outcome or need for treatment on intensive care unit were regarded as moderate manifestations of COVID-19. Admission to ICU was indicated by the treating physicians in case of poor oxygenation despite supplemental oxygen or secondary organ failure. According to our hospital COVID-19 protocol poor oxygenation was defined as respiratory rate> 30/min and oxygen saturation <90% despite 8 l supplemental oxygen via mask. Due to compensated ICU capacities there was no delay in admission to ICU or indication for triage. Each patient without an event had at least 40 days of follow-up.

### Data collection

Demographic, clinical, treatment, and outcome data were reviewed following a standardized data collection protocol from the electronic medical records by individual chart review. Due to inclusion of critical ill patients from external hospitals after transfer to our hospital for ICU treatment a complete information regarding medical history or inflammatory markers at onset was not obtainable. We evaluated hospital admission, duration of hospitalization, transfer to intensive care unit, duration of intensive care unit, discharge from hospital and death.

### Chest CT-imaging and analysis of CAC

All examinations were performed on a 64- or 256-slice CT system (IQon or iCT, respectively; Philips Healthcare, the Netherlands). The slice thickness was 3mm, the reconstruction algorithm was F: YB. CAC was calculated on standard CT-scans of the thorax using a dedicated post-processing software (Philips Intellispace Portal 10.0; Philips Healthcare) with the Agatston method [19].

### Statistical analysis

Categorical variables were expressed as frequencies and percentages, continuous variables were expressed as means ± standard deviation or as median (inter-quartile range [IQR]). All statistical evaluations are based on the event-free survival for the study endpoints using the

Kaplan-Meier method; hazard ratios (above the median) were calculated with the Cox proportional hazard model. Further, odds ratios were calculated using logistic regression. Statistical significance was assumed for two-sided P-values <0.05. The statistical package R, version 3.5.3 including the package rms and EpiR was used for analysis [20].

## Results

### Patient population

We included 109 patients with laboratory confirmed diagnosis of SARS-CoV-2 in this study. Mean age in our study population was 61.8 ± 17.5 years. 74 (67.9%) were male. In our study population 48 patients needed ICU treatment. We observed 11 deaths due to terminal lung or multi organ failure. The composite endpoint occurred in 48 patients. Baseline characteristics are shown in Table 1.

### CT-analysis

Mean CAC on chest CT was 753. The median of CAC was 31[IQR: 0–557]. A CAC of 0 was observed in 40 (36.7%) patients. A CAC of 0 could be observed in 47.6% patients with moderate, in 22.9% of patients with critical outcome and in 18.2% in patients with fatal outcome. In patients with critical outcome median CAC was 140[IQR 1–1165], in patients with fatal outcome CAC was 160[IQR: 88–562]. A quantification of calcified plaque burden in the native COVID-19 screening CT in two representative patients is depicted in Fig 1.

### Outcome

61 (55.9%) patients had a moderate outcome and were hospitalized on normal ward. 48 (44.4%) patients had a critical outcome and were treated on ICU. 11 (10.1%) patients had a fatal outcome and died from respiratory failure or multi-organ failure.

In patients with CAC above the group's median we observed significantly more events regarding the critical outcome (HR: 1.97[1.09,3.57], p = 0.026), the fatal outcome (HR: 4.95 [1.07,22.9], p = 0.041) and the composite endpoint (HR: 2.31[1.28,4.17], p = 0.0056). Kaplan-Meier-curve for the composite endpoint is provided in Fig 2; p for Log-Rank was 0.003.

Also, in patients with a CAC above the median the risk was significantly increased for critical outcome (OR: 3.01 [1.37, 6.61], p = 0.01), for fatal outcome (OR: 5.3 [1.09, 25.8], p = 0.02), and for composite endpoint (OR: 3.01 [1.37, 6.61], p = 0.01) as depicted in Table 2. Prolonged and severe course of COVID-19 with ICU treatment for more than 14 days and death was

**Table 1. Patient characteristics according to the presence of study endpoints.**

|  | all patients | moderate | critical | fatal |
|---|---|---|---|---|
| **patients, n** | 109 | 61 | 48 | 11 |
| **mean age, yrs. (SD)** | 61.8 ± 17.5 | 59.0±18.0 | 65.8±16.0 | 69.2±18.0 |
| **male (%)** | 74 (67.9) | 35 (57.4) | 39 (81.3) | 8 (72.7) |
| **median ICU, days (range)** | 16[9–42] | 0 | 16[9–42] | 16[13–18] |
| **median CAC (IQR)** | 31[0–557] | 6[0–218] | 140[1–1165] | 160[88–562] |
| **patients w/ CAC = 0, n (%)** | 40 (36.7) | 29 (47.6) | 11 (22.9) | 2 (18.2) |

Moderate, hospitalized COVID-19 patients. Critical, COVID19 patients transferred to intensive care unit (ICU). Fatal, deceased COVID-19 patients. Patients with fatal outcome are a subgroup of patients with critical outcome due to transfer to ICU. All patients with fatal outcome were patients on ICU. CAC, coronary artery calcium score.

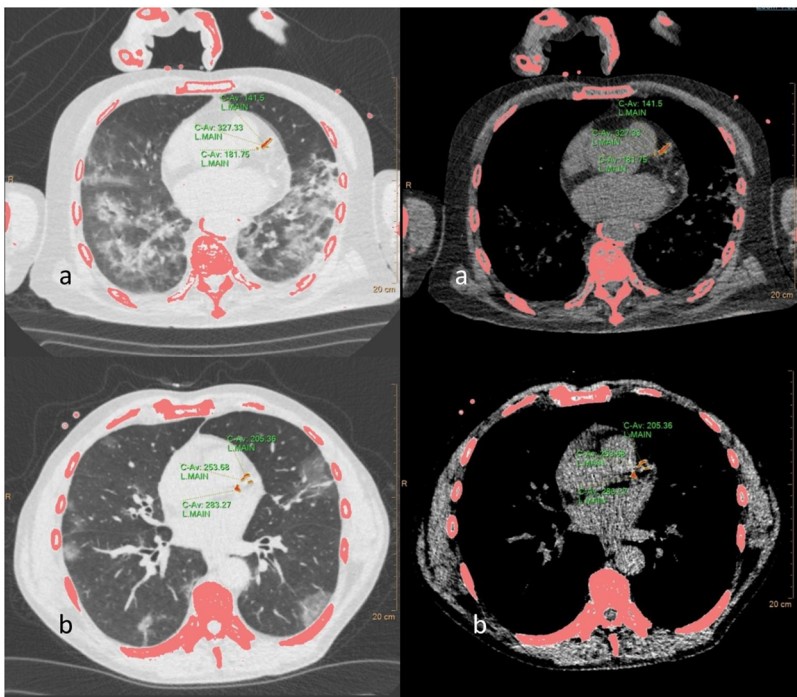

**Fig 1. Quantification of calcified plaque burden in the LAD in the native COVID-19 screening CT in two representative patients a) 70 yrs, male b) 53 yrs, male.** Both patients had a prolonged period >14 days on intensive care unit.

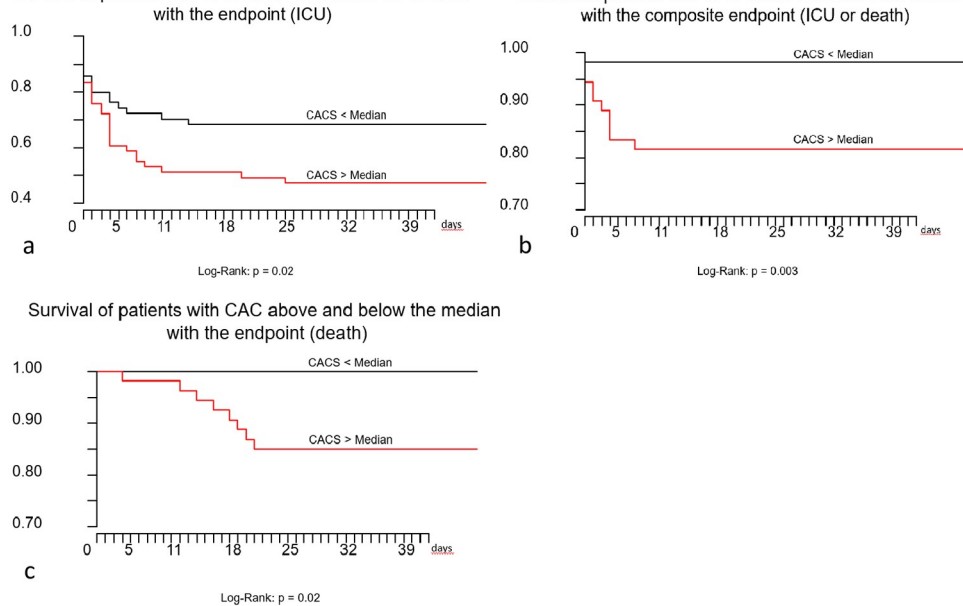

**Fig 2. Kaplan-Meier-curves for critical outcome (a), for composite endpoint death or intensive care unit treatment (b) and fatal outcome (c).** A significant reduction of event free survival could be shown for the critical outcome (p = 0.02), composite endpoint (p = 0.003) and death (p = 0.02).

**Table 2. Odds ratios in COVID-19 patients for comparison of patients with CAC above the median versus patients with CAC below the median.**

|  | OR [95% CI] | p-value |
|---|---|---|
| fatal outcome (death) | 5.3 [1.09, 25.8] | 0.02 |
| critical outcome (admission on ICU) | 3.01 [1.37, 6.61] | 0.01 |
| critical and fatal outcome | 3.01 [1.37, 6.61] | 0.01 |
| prolonged ICU treatment (>14d) and death | 2.56 (1.03, 6.36) | 0.04 |

OR with the endpoints critical outcome (admission on ICU) and composite endpoint critical and fatal outcome were similar due to fatal outcome after transfer to ICU.

significantly more often observed in patients with CAC above the median (OR: 2.56 [1.03, 6.36], p = 0.04).

Male gender was adversely associated with need for treatment on ICU (HR: 2.34[1.13,4.85], p = 0.022). Age and male gender were adversely associated with the composite endpoint (HR: 1.82[1.15,2.89], p = 0.01 and HR: 2.4[1.16,4.96], p = 0.018) and not with the fatal outcome (HR: 2.12[0.79,5.73], p = 0.14 and HR: 1.27[0.34,4.77], p = 0.73), respectively.

## Discussion

The major findings of this study were that CAC may be useful as an additional risk predictor for critical and fatal outcome of COVID-19 and that a CAC above the median is associated with a higher risk for treatment on intensive care unit and mortality.

COVID-19 causes in a substantial number of patients a distinct damage of the lungs and consequently leads to respiratory failure. As a consequence of the infection and the excessive inflammatory reaction, COVID-19 may involve other organ systems and may lead to severe and persistent organ injury. Despite intensive care treatment many patients die due to respiratory and secondary organ failure.

The underlying factors for COVID-19 severity are poorly understood yet and so far there are only moderately effective treatment options [21]. Hence, the identification of risk factors and risk populations for severe outcome is of particular interest to reduce further serious impact on the population. Recent epidemiological studies have shown various risk factors that may indicate a critical or fatal outcome in patients with COVID-19 [3, 6, 10–12, 15, 22]. Particularly, cardiovascular risk factors (e.g. arterial hypertension or diabetes) and CAD were identified as potential risk factors for severe outcome. Whereby other factors such as age may have an influence on the relationship between COVID-19 severity and CAD as reported in recent studies [3, 10, 11, 22, 23].

Evidence suggest that CAD is adverse correlated with clinical outcome in COVID-19. However, current studies simply used information based on the medical history of the included patients [10, 12, 14, 24]. However, there is no information on pathological or radiological results regarding CAD presence and therefore medical history may underestimated the proportion of CAD on this population and patients with unknown CAD may be missed by the medical history only. These patients with subclinical CAD could be theoretically identified by CAC measuring provided by the COVID-19 chest CT, a meanwhile widely accepted screening method derived from the initial experience which we have implemented in our daily routine [7, 8].

We conducted this retrospective study to gain further insights into the impact of CAD as a risk factor for COVID-19 severity. To our knowledge, our study is the first investigating the topic of CAC as marker of CAD and its prognostic value for worse clinical outcome in

COVID-19. Our study elucidates the relationship of CAD and COVID-19 for severe outcome and whether an additional parameter can be identified at the time of hospital admission in order to identify a risk population and thus reserve intensive care capacities at an early stage in case of clinical deterioration. In our population we have demonstrated that a coronary calcium score above 31 may be suitable for the prediction of adverse outcome associated with COVID-19.

We used CAC as a widely accepted marker for CAD. CAC can be easily calculated from native computed tomography scans and therefore simultaneously be acquired with imaging for COVID-19. As previously shown, CAC offers non-invasively information as a risk predictor regarding CAD even in patients before symptomatic CAD is diagnosed [16, 17].

Our data shows a reduced survival in patients with a CAC above the median, reflecting an association between CAD and COVID-19 outcome. A CAC above the median reveals a significantly increased risk for critical and fatal outcome. We found a significant association between elevated CAC and need for treatment on intensive care unit and fatal outcome. In addition, a prolonged intensive care treatment and death were associated with an increased CAC.

The elevated risk for patients with CAD in COVID-19 is assumed to be related to differential mechanism of injury, including damage of the microvasculature, endothelial shedding, hypoxia-induced myocardial injury which may be aggravated by relative ischemia [25, 26]. Autopsy data suggest that SARS–CoV-2 may spread via the bloodstream and infect the heart [27]. Patients with CAD and endothelial dysfunction may have a severe course due to relative ischemia under inflammatory stress or direct infection [25].

Elevated Troponin levels, reflecting myocardial damage in COVID-19, were associated with severe outcome [15]. COVID-19 often leads to cardiac injury and patients with CAD may be more susceptible for this injury [10, 11, 13, 14, 24]. In addition, a cytokine/inflammation-mediated damage is also very likely in cardiac injury and endothelial dysfunction may play a role [25].

Our data show significant association for a critical outcome and fatal outcome in patients with COVID-19. Therefore, our data confirms results from earlier studies showing CAD, as a diagnosis based on medical history, is a risk factor for severe outcome in COVID-19. One advantage of the determination of CAC in COVID-19 screening CT is that it also includes so far undiagnosed patients with CAD. As a result, patients with an increased risk can be identified more precisely. CAC screening can therefore lead to a more accurate risk stratification even in patients who would not have been identified previously if the risks had been determined solely based on the medical history.

The main limitations of the study are the retrospective single-center study design and the small sample size. Another limiting factor is that only hospitalized patients were included and therefore no data regarding mild cases of COVID-19 could be obtained. However, our study is conducted on one of the largest single center cohort of COVID-19 in Germany. By including hospitalized cases with moderate to severe COVID-19 our data has a high case fatality rate compared to other studies. Another limitation is the use of non-gated fast-helical chest CT scans for quantification of coronary artery calcification. Chest CT in our patients was performed to either rule out COVID-19 pneumonia or to evaluate the extent of the infection. The chest CT on admission was not intended for analysis of CAC and therefore not ECG-gated. However, there are publications showing that non-gated fast-helical chest CT can provide reliable and accurate calcium scoring in coronary arteries [28–30].

Our study is the first study investigating the topic of CAC as marker and risk predictor of coronary artery disease and its prognostic value for COVID-19 outcome. A major effect of age on the prognostic value of CAC regarding survival and occurrence of events is very likely. Confirmation of the study findings should be performed in larger populations, also longitudinal studies should be performed. In particular, a correlation of CAC to inflammatory parameters may provide further information for risk assessment.

## Conclusion

CAD is a common condition among patients hospitalized with COVID-19 and is associated with a higher risk for treatment on intensive care unit and mortality in comparison to patients without cardiovascular affection. Although the underlying mechanism of this association needs to be further elucidated, our findings highlight the potential of obtaining CAC during COVID-19 chest CT-screening as a radiologic marker for CAD providing a general risk assessment for patients hospitalized for COVID-19 infection beyond those which have been described so far in the literature. Thus, CAC may be useful in predicting critical and fatal outcome of COVID-19.

## Supporting information

**S1 Data.**
(XLSX)

## Acknowledgments

The authors want to thank the COVID-19 team in our hospital.

## Author Contributions

**Conceptualization:** Gregor S. Zimmermann, Alexander A. Fingerle, Jonathan Nadjiri.

**Data curation:** Alexander A. Fingerle, Christina Müller-Leisse, Felix Gassert, Claudio E. von Schacky, Bernhard Haller, Jonathan Nadjiri.

**Formal analysis:** Gregor S. Zimmermann, Alexander A. Fingerle, Christina Müller-Leisse, Tareq Ibrahim, Christoph Spinner, Bernhard Haller, Jonathan Nadjiri.

**Methodology:** Tareq Ibrahim, Bernhard Haller.

**Writing – original draft:** Gregor S. Zimmermann, Alexander A. Fingerle.

**Writing – review & editing:** Gregor S. Zimmermann, Alexander A. Fingerle, Tareq Ibrahim, Karl-Ludwig Laugwitz, Fabian Geisler, Christoph Spinner, Markus R. Makowski, Jonathan Nadjiri.

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
