## [Decision Letter · Decision Letter 0]

17 Nov 2020

PONE-D-20-29515

Submitting Original Article to PLOS ONE

Coronary Calcium Scoring assessed on native screening chest CT imaging as predictor for outcome in COVID-19: an analysis of a hospitalized german cohort.

PLOS ONE

Dear Dr. Zimmermann,

Thank you for submitting your manuscript to PLOS ONE. After careful consideration, we feel that it has merit but does not fully meet PLOS ONE’s publication criteria as it currently stands. Therefore, we invite you to submit a revised version of the manuscript that addresses the points raised during the review process.

We look forward to receiving your revised manuscript.

Kind regards,

Andreas Zirlik, MD

Academic Editor

PLOS ONE

Journal Requirements:

2. In ethics statement in the manuscript and in the online submission form, please provide additional information about the patient records used in your retrospective study. Specifically, please ensure that you have discussed whether all data were fully anonymized before you accessed them and/or whether the IRB or ethics committee waived the requirement for informed consent. If patients provided informed written consent to have data from their medical records used in research, please include this information.

3. During your revisions, please note that a simple title correction is required, as some stock text has been left in the online submission form. The title should be "Coronary Calcium Scoring assessed on native screening chest CT imaging as predictor for outcome in COVID-19: an analysis of a hospitalized German cohort". Please ensure this is updated in the manuscript file and the online submission information.

4.We note that you have indicated that data from this study are available upon request. PLOS only allows data to be available upon request if there are legal or ethical restrictions on sharing data publicly. For information on unacceptable data access restrictions, please see http://journals.plos.org/plosone/s/data-availability#loc-unacceptable-data-access-restrictions.

5. Please include your tables as part of your main manuscript and remove the individual files. Please note that supplementary tables (should remain/ be uploaded) as separate "supporting information" files

Additional Editor Comments (if provided):

This is a straight forward and interesting analysis describing an association between CAC and outcome of COVID 19 patients.

Any correlation with infalmmatory markers?

Reviewers' comments:

Reviewer's Responses to Questions

**Comments to the Author**

1. Is the manuscript technically sound, and do the data support the conclusions?

Reviewer #1: Yes

2. Has the statistical analysis been performed appropriately and rigorously? 

Reviewer #1: Yes

3. Have the authors made all data underlying the findings in their manuscript fully available?

Reviewer #1: Yes

4. Is the manuscript presented in an intelligible fashion and written in standard English?

Reviewer #1: Yes

5. Review Comments to the Author

Reviewer #1: The authors investigated the role of CAC in predicting the risk of ICU admission and complications/mortality. The manuscript is well and clearly written, and the conclusions are sound.

Comments follow below.

1) I assume that patient data were obtained by individual chart review, please describe this in the methods. If so, the paper would benefit from additional information on comorbidities of the included patients that have previously been shown to adversely affect the course of COVID-19, such as known CAD, hypertension, diabetes, obesity, and others. An additional table showing these patient characteristics would be appropriate.

2) ICU transfer has been chosen as an endpoint. Both reasons for admission to ICU and complications/course of treatment during ICU stay can vary quite a bit. Is any more information available on ICU admission reasons (only dyspnea or other reasons/accompanying diseases; admissions because of disease severity versus availability of ICU beds to be covered; how was ICU bed capacity; etc.) or on the course of treatment (intubation necessary or highflow oxygenation sufficient, complications, etc.). Providing more detailed information may help to appreciate the usefulness of CAC as a predictive parameter.

6. PLOS authors have the option to publish the peer review history of their article (what does this mean?). If published, this will include your full peer review and any attached files.

Reviewer #1: No

---

## [Author Response · Author response to Decision Letter 0]

20 Nov 2020

Submitting Original Article to PLOS ONE

Title: “Coronary Calcium Scoring assessed on native screening chest CT imaging as predictor for outcome in COVID-19: an analysis of a hospitalized German cohort”

Authors:

Gregor S. Zimmermann, Alexander Fingerle, Christina Müller-Leisse, Felix Gassert, Claudio E. von Schacky, Tareq Ibrahim, Karl-Ludwig Laugwitz, Fabian Geisler, Christoph Spinner, Bernhard Haller, Markus R. Makowski and Jonathan Nadjiri 

Dear Editor,

Enclosed you will find the revised version of our manuscript “Coronary Calcium Scoring assessed on native screening chest CT imaging as predictor for outcome in COVID-19: an analysis of a hospitalized German cohort”, which we would like to resubmit to PLOS ONE.

We have addressed each of the journal requirements, the editors ‘comments and the review comments to the author. By responding to the detailed and thoughtful comments of the reviewer, the paper has been substantially improved. Changes in the manuscript are marked.

Thank you very much for reconsidering our manuscript for publication in PLOS ONE.

With best regards,

Gregor S. Zimmermann, M.D.

Department of Internal Medicine I, Division of Respiratory Diseases

Klinikum rechts der Isar, Technical University Munich

Ismaninger Str. 22

81675 Munich, Germany

gregor.zimmermann@tum.de

We thank the reviewers for your helpful comments. By addressing your suggestions, the manuscript has been significantly improved. We marked changes in the text. The comments have been addressed as follows:

Journal Requirements:

1. We have checked whether the manuscript meets the requirements of PLOS ONE.

2. We provided additional information about the patient’s records reviewed for our retrospective study. The IRB waived the requirements for informed consent and we added this important information to the manuscript. Data acquisition and processing has been approved for this particular study by the local ethics committee.

3. We corrected the title as suggested. 

4. a) Following local laws and Clinical Institutional Review Board the data sharing is restricted due to potentially identifying patient information in this study. We added a excel sheet of the CAC and the endpoints of our study. An identification of an individual patient is not possible in this excel sheet. A reasonable request for a data access could be sent by researchers who meet the criteria for access to confidential data to the corresponding author only after permission from the Clinical Institutional Review Board (Director: Professor Dr. G. Schmidt, contact: ethikkommission@mri.tum.de).

b) We uploaded a minimal anonymized data set as supporting information file.

5. Tables were included as part of the manuscript and the individual files were removed. 

6. We corrected this and included our ethics statement in the methods section and deleted it from another section. 

Answer to Additional Editors Comments:

The editors raised an interesting question of any correlation with inflammatory markers. From the perspective of a tertiary medical centre many patients were transferred from external hospitals after several days for intensive care treatment, so comparable time points of inflammatory markers couldn´t be established. However, the editor added a valuable comment and we included this point in the discussion as an important goal of further investigation. 

Review Comments to the Author:

1) We added a more detailed description of individual chart review in the methods section. Due to transfer of sedated and intubated patients from external hospitals to our tertiary medical centre a complete data set of the medical history was not obtainable in all cases. The main purpose of our study was to identify a parameter to predict the individual patients‘ risk with COVID-19 independently from clinical information and medical history by a CT scan on admission. 

2) We added more precise information regarding ICU admission reasons in the method´s section. Our hospital has established a clearly defined protocol for admission on ICU. Admission to ICU was indicated by the treating physicians in case of poor oxygenation despite supplemental oxygen or secondary organ failure. According to our hospital COVID-19 protocol poor oxygenation was defined as respiratory rate> 30/min and oxygen saturation <90% despite 8 liter/minute supplemental oxygen via mask. Due to compensated ICU capacities there was no delay in admission to ICU or indication for triage. We added this information in the methods section.

Thank you very much for receiving our manuscript and considering it for review. The editors and reviewers comments have overall augmented the value of our paper. We hope that our manuscript meets your high scientific standards and will be eligible for publication in PLOS ONE. We appreciate your time and look forward to your response.

Sincerely,

Gregor Zimmermann, M.D.

Department of Internal Medicine I

Klinikum rechts der Isar

Technical University Munich

Ismaninger Str. 22

81675 Munich, Germany

Tel.: +49 89 41405803, Fax.: +49 89 4140 4362

gregor.zimmermann@tum.de

---

## [Editor Report · Decision Letter 1]

16 Dec 2020

Coronary Calcium Scoring assessed on native screening chest CT imaging as predictor for outcome in COVID-19: an analysis of a hospitalized German cohort.

PONE-D-20-29515R1

Dear Dr. Zimmermann,

We’re pleased to inform you that your manuscript has been judged scientifically suitable for publication and will be formally accepted for publication once it meets all outstanding technical requirements.

Kind regards,

Andreas Zirlik, MD

Academic Editor

PLOS ONE

Additional Editor Comments (optional):

The authors adequately addressed the concerns of Reviewer and Editor.
---

## [Editor Report · Acceptance letter]

18 Dec 2020

PONE-D-20-29515R1 

Coronary Calcium Scoring assessed on native screening chest CT imaging as predictor for outcome in COVID-19: an analysis of a hospitalized German cohort 

Dear Dr. Zimmermann:

I'm pleased to inform you that your manuscript has been deemed suitable for publication in PLOS ONE. Congratulations! Your manuscript is now with our production department. 

Kind regards, 

on behalf of

Univ. Prof. Dr. Andreas Zirlik 

Academic Editor

PLOS ONE